# Cell Type-Specific Patterns in the Accumulation of DNA Damage Following Multifractional Radiation Exposure

**DOI:** 10.3390/ijms232112861

**Published:** 2022-10-25

**Authors:** Pamela Akuwudike, Adrianna Tartas, Milagrosa López-Riego, Iuliana Toma-Dasu, Andrzej Wojcik, Lovisa Lundholm

**Affiliations:** 1Centre for Radiation Protection Research, Department of Molecular Biosciences, The Wenner-Gren Institute, Stockholm University, 10691 Stockholm, Sweden; 2Biomedical Physics Division, Faculty of Physics, Institute of Experimental Physics, University of Warsaw, 02-093 Warsaw, Poland; 3Department of Oncology-Pathology, Karolinska Institutet, 17176 Stockholm, Sweden; 4Department of Physics, Stockholm University, 10691 Stockholm, Sweden; 5Institute of Biology, Jan Kochanowski University, 25-369 Kielce, Poland

**Keywords:** second malignant neoplasms (SMN), multifractionated radiation exposure, micronuclei, DNA damage, radiotherapy, giant nuclei, nuclear buds, residual DNA damage

## Abstract

Predicting the risk of second malignant neoplasms is complicated by uncertainties regarding the shape of the dose–response relationship at high doses. Limited understanding of the competitive relationship between cell killing and the accumulation of DNA lesions at high doses, as well as the effects of other modulatory factors unique to radiation exposure during radiotherapy, such as dose heterogeneity across normal tissue and dose fractionation, contribute to these uncertainties. The aim of this study was to analyze the impact of fractionated irradiations on two cell systems, focusing on the endpoints relevant for cancer induction. To simulate the heterogeneous dose distribution across normal tissue during radiotherapy, exponentially growing VH10 fibroblasts and AHH-1 lymphoblasts were irradiated with 9 and 12 fractions (VH10) and 10 fractions (AHH-1) at 0.25, 0.5, 1, or 2 Gy per fraction. The effects on cell growth, cell survival, radiosensitivity and the accumulation of residual DNA damage lesions were analyzed as functions of dose per fraction and the total absorbed dose. Residual γH2AX foci and other DNA damage markers (micronuclei, nuclear buds, and giant nuclei) were accumulated at high doses in both cell types, but in a cell type-dependent manner. The competitive relationship between cell killing and the accumulation of carcinogenic DNA damage following multifractional radiation exposure is cell type-specific.

## 1. Introduction

Advances in cancer therapy technologies along with supportive post-therapy healthcare have contributed to a steady rise in the 5-year survival of cancer patients [1]. However, the ionizing radiation that is used to treat approximately 50% of cancers is a double-edged sword in that it is also a carcinogen itself [2]. Thus, the improved survival comes at the cost of an elevated risk of developing radiation-induced second malignant neoplasms (SMNs), especially when treatment takes place during childhood and adolescence [3,4,5,6,7,8]. SMN risk could be included as a factor in the process of treatment planning optimization, but to this end the dose–response relationship must be known. The estimation of SMN risk by epidemiological approaches following radiotherapy is not straightforward because radiation-induced cancers appear after a long latency period. Thus, the SMN seen today result from radiotherapy treatments carried out many years ago, often with incomplete information of dose distribution in healthy tissue surrounding the tumor [9].

Predictive mathematical models could circumvent drawbacks encountered during epidemiological studies, for example, by accounting for new modulatory factors like the constant evolution of treatment modalities [10], and the lengthy follow-ups due to the latency of second malignancies [4]. However, developing and validating predictive models is often restricted by the uncertainties associated with the shape of the dose–response curve for radiation-induced SMNs at high doses (clinically relevant doses) [11,12]. Studies involving dose reconstruction and retrospective dosimetry have shown that a high percentage of radiation-induced SMNs occur within a radiation field where doses are high [13,14]. However, it is still unclear if the incidence of SMNs increases linearly with increasing dose, drops off after a threshold dose, or remains constant beyond a specific dose [10,12,15].

Based on curated epidemiological data, three models attempt to describe the dose–response relationship at high doses. The linear no-threshold model (LNT) is based on the experiences of the atomic bomb survivors and is regarded as the gold standard for estimating cancer risk at low doses [16]. Atomic bomb survivors experienced a single, acute whole-body radiation exposure, with a significant linear increase in cancer risk from doses ranging from 0.1–2 Gy [16]. In comparison, patients receiving radiotherapy are exposed to much higher doses, often greater than 50 Gy, delivered typically in daily fractions (or rarely, as multiple fractions per day), over a period of weeks in small body volumes. Extrapolating cancer risk for these patients based on the LNT model is unsuitable due to the differences in exposure scenarios and the uncertainties associated with doses above 2 Gy [12,17]. Despite this discrepancy, the incidence of some cancer types, such as breast cancer in childhood survivors of Hodgkin’s diseases, has been shown to increase linearly with dose [15,18]. It has long been assumed that increased cell killing at high doses reduces the population of cells with radiation-induced mutations. This is the mechanistic basis for the competition model (also known as the cell sterilization model), which is a modification of Gray’s bell-shaped model [19]. Although the competition model is evident in the incidence of thyroid cancer, lung cancer, and in vitro studies using Chinese hamster CH3 10T1/2 cells [20], this model does not fit the cancer incidence for other cancer types. The cell sterilization model has been criticized by Sachs and Brenner [21], who proposed a plateau model, which accounts for cell killing, induction of carcinogenic mutation, and cell proliferation or repopulation. Uncertainties remain, predominantly due to knowledge gaps concerning the roles of factors unique to radiation exposure at high doses, such as the sparing effect of dose fractionation [22] and the impact of heterogeneous dose distribution across normal tissue [2]. Furthermore, there is a limited understanding of biological effects at high doses, such as the competitive relationship between neoplastic transformation of irradiated cells and cell killing, and whether cell repopulation maintains a steady state of transformations over a certain dose threshold.

Radiation-induced DNA damage is a major mechanism of injury incurred by normal tissue during radiotherapy [23]. DNA double-strand breaks (DSBs) are potentially lethal DNA lesions capable of triggering cell death pathways, such as apoptosis and senescence [24]. However, aberrant repair of DSBs is associated with numerous markers of genomic instability. Examples include micronucleus (MN) formation, indicating a loss of genomic information, commonly of an acentric chromosomal fragment after radiation exposure [25], and translocations, indicating stable chromosomal rearrangements [26]. DSBs are also associated with formation of nuclear buds, which serve as indicators of gene amplification (of oncogenes) [27,28]. The formation of large mononucleated cells (giant nuclei, GN) is evidence of polyploidy as a result of cell cycle arrest and the onset of an endoreplication cycle, during which cells replicate their genomes without cell division [29]. This stress-induced polyploidy has been shown to occur in p53-deficient cells [30] and is implicated in cancer metastasis in tumors [29,31]. Residual γH2AX foci represent unrepaired DNA double-strand breaks and are considered to be markers of late radiation toxicity. All these markers of genomic instability have been implicated in neoplastic transformations and are evidence of the fact that the resolution of DSBs plays a pivotal role in cell fate decisions. The accumulation of DSBs activates various cell-killing mechanisms but can also be the initiating event in the formation of genomic instability and the accumulation of mutations [32]. Therefore, determining the competitive relationship between cell killing and the accumulation of genomic instability in normal cells after fractionated radiation exposure requires an investigation into the biological effects of fractionated radiation exposure at multiple dose levels on the accumulation of DNA DSBs and the different types of chromosomal damage associated with aberrant resolution of DBSs.

We set up an in vitro study to investigate the biological effects of heterogeneous dose distribution on the competition between cell killing and accumulation of DNA damage in surviving cells. We designed fractionation schemes tailored to two human normal cell types (VH10 fibroblasts and AHH-1 lymphoblasts), representing elastic and hierarchical tissues, respectively. VH10 fibroblasts are normal primary cells with a doubling time of 36–40 h; these cells are relatively radioresistant, with a lethal dose to 50% (LD50) of 2 Gy. AHH-1 cells are more radiosensitive (LD50: 0.5 Gy), with a doubling time of 16–19 h. VH10 fibroblasts are adherent cells of mesenchymal origin, and their programmed cell death pathway is senescence [33]. AHH-1 lymphoblasts are of non-adherent hematopoietic cell type, and their cell death mode is apoptosis [34]. To simulate the non-uniform dose distribution across normal tissue, we irradiated both cell types at different doses per fraction. These ranged from 2 (planned clinical dose per fraction), to 1 (high dose) and 0.5 (moderate dose), to 0.25 (low–moderate dose) Gy per fraction (Figure 1). Total absorbed doses are summarized in Table 1. We determined the effects on cell growth, DNA damage, cell survival, and radiosensitivity as endpoints of both cell death and induction of genomic instability. Our study highlights the cell type-specific differences in the accumulation of DNA damage in normal cells after fractionated radiation exposure at radiotherapy-relevant doses, which was not described previously to any larger extent.

## 2. Results

### 2.1. Delayed Growth of Cells Irradiated at High Doses-per-Fraction

Cell growth was monitored weekly during and post-fractionated radiation exposure (PFRE). PFRE and VH10 fibroblasts were passaged for 30 days at 10-day intervals, while AHH-1 lymphoblasts were passaged three times a week for 21 days. During fractionated radiation exposure (DFRE), there was a dose-per-fraction-dependent decline in the slope of the cumulative population doubling (PD) curve (Figure 2A, Appendix A) in both cell lines. Growth arrest, demonstrated by a horizontal PD curve, was observed in both cell types irradiated with 2 Gy per fraction. However, the population of AHH-1 cells reduced in number, i.e., had negative PD values, towards the end of radiation exposure. PFRE, the PD curves of VH10 and AHH-1 cells exposed to 0.25 and 0.5 Gy per fraction were not significantly different from the non-irradiated control and each other. The cell growth characteristics of both cell types only changed at high doses-per-fraction, with a lag in cell growth at 1 Gy/fraction and a prolonged cell growth arrest at 2 Gy/fraction (Figure 2B). Growth arrest (9 fractions) or negative population doubling (12 fractions) persisted in VH10 irradiated at 2 Gy per fraction for 15–20 days PFRE. In AHH-1 cells, a loss in cell population indicated by a negative population doubling was observed during the first 10 days PFRE, with a subsequent recovery.

### 2.2. Excess DNA Damage at High Doses

Next, we investigated the effects of multifractional radiation exposure at different dose levels on the accumulation of residual γH2AX foci, measured 3 and 7 days PFRE. Raw values of focus frequency per cell are shown in Figure 3A as a function of dose levels. Based on focus size distribution, radiation-induced foci (total foci, TF) could be classified into either small foci (SF) or large foci (LF) to distinguish simple DNA damage lesions from more complex DNA lesions, respectively [35]. A general trend of increased residual focus frequencies at 3 days PFRE was observed in VH10 (after 9 fractions and 12 fractions) and AHH-1 cells irradiated with 1 or 2 Gy/fraction (Figure 3A). Seven days PFRE, focus frequency was significantly higher in 2 Gy/fraction in VH10 fibroblasts (after 9 fractions) compared to cells irradiated at 0.25 and 0.5 Gy/fraction (Figure 3A). Similar patterns were observed in the accumulation of SF (Figure 3A). The large focus (LF) frequency increased significantly 3 days PFRE in VH10 fibroblasts (after 9 fractions) irradiated at the 2 Gy/fraction compared to the unirradiated controls (Figure 3A). Although not statistically significant, declining focus frequency in VH10 fibroblasts after 9 fractions was detected in cells irradiated at 0.25 and 0.5 Gy/fraction 7 days PFRE. On the contrary, the decline in focus frequency in AHH-1 lymphoblasts was present in cells irradiated at 1 and 2 Gy/fraction (Figure 3A). Focus frequencies in VH10 fibroblasts after 12 fractions were generally lower compared to cells given 9 fractions (Figure 3A).

Dose–response relationships for residual γ-H2AX foci as a function of total absorbed dose (Table 1) were derived using net focus frequency (the control values subtracted from the average foci per cell for each dose per fraction) at both time points, where negative values denote a depletion in focus frequency compared to the control, and positive values represent an excess. Net foci values show a trend for excess foci frequency at doses above 10 Gy at 3 days PFRE in both cell types, regardless of the number of fractions. Temporal differences in the dose–response relationship generated with residual γH2AX were observed. Net foci values 3 days PFRE were fit to a linear relationship, while a linear–quadratic relationship was used for 7 days PFRE values (Figure 3B). The decline in residual γH2AX foci at lower doses in VH10 fibroblasts 7 days PFRE and at high doses in AHH-1 lymphoblasts was evident in the dip of the dose–response curves.

### 2.3. Cell Type-Specific Differences in the Accumulation of Genomic Instability

A high frequency of micronuclei (MN) was observed PFRE during the scoring of irradiation-induced foci, and then quantified. Other genomic alterations were preferentially present in separate cell types, and thus nuclear buds (NBD) were scored in VH10 fibroblasts and giant were mononucleated nuclei (GN) quantified in AHH-1 lymphoblasts (Figure 4A). The frequencies of MN, NBD, and GN in both cell types as a function of dose/fraction are summarized in Figure 4B,C. A dose-dependent increase in MN and NBD frequency was observed in VH10 fibroblasts (9 and 12 fractions) 3 days PFRE at 0.25–2 Gy/fraction, yet MN frequency was plateauing above 1 Gy/fraction for 12 fractions (Figure 4B). However, in AHH-1 lymphoblasts, MN as well as GN frequency (Figure 4C) were only significantly higher than the control in cells irradiated at 2 Gy/fraction.

Dose–response curves for MN and GN frequency were fitted to a linear quadratic model (Figure 5A). MN frequency in VH10 fibroblasts increased linearly until it peaked at a total dose of 15 Gy (after 9 fractions) and 20 Gy (after 12 fractions), while AHH-1 lymphoblasts had an exponential increase in MN frequency at a total dose above 5 Gy. Similar results were observed for the dose–response relationships plotted for GNs. NBD frequency in VH10 fibroblasts was fitted to a linear equation (Figure 5A). To further demonstrate these dose–response relationships, we carried out simple simulations using the fitting coefficients summarized in the Appendix A, derived from our fits of 3 days PFRE, and further extended the total absorbed dose to 50 Gy. This corresponds to 25 fractions of 2 Gy, or for a hypo-fractionated scheme, 5.5 Gy/fraction and 4.16 Gy/fraction in VH10 fibroblasts after 9 and 12 fractions respectively, and 5 Gy/fraction in AHH-1 lymphoblasts (Figure 5B). A close similarity between simulated and experimental data was observed for MN in both cell types and GN frequencies in AHH-1 cells. However, these simulations further demonstrate cell type-specific differences in the accumulation of MN, NBD, and GN with increasing dose, with a bell-shaped dose–response for MN in VH10 cells, and an exponential response in AHH-1 lymphoblasts. NBD frequency also had a linear relationship with increasing dose.

### 2.4. Differential Pattern in Cell Survival and Radiosensitivity

To determine the delayed effect of fractionated radiation exposure on the ability of cells to form colonies, we carried out a colony-forming assay 10 days PFRE (Figure 6A). A dose-per-fraction and total dose-dependent decrease in colony formation was observed in VH10 fibroblasts. However, colony formation in AHH-1 lymphoblasts did not decline at any dose level, except at a total dose of 20 Gy (2 Gy/fraction). Instead, there was a trend towards an increase of 2.5–10 Gy. To determine the effect of fractionated radiation exposure on the radiosensitivity of cells 20 days PFRE, surviving cells previously exposed to 0.25, 0.5 and 1 Gy per fraction were exposed to increasing, acute doses of radiation, and clonogenic cell survival was determined (Figure 6B). VH10 fibroblasts were exposed to 2–8 Gy, and AHH-1 lymphoblasts to 1–4 Gy, in order to account for their different radiosensitivity. Comparison of the survival curves did not indicate any significant changes in the radiosensitivity of VH10 fibroblasts, which were previously given 9 fractions (Figure 6B), compared to the control cells. However, a clear trend towards an acquired increase in the radioresistance was observed in AHH-1 lymphoblasts by increasing doses per fraction.

## 3. Discussion

This study was designed as an in vitro experiment to simulate fractionated radiation exposure and dose inhomogeneity across normal tissues, as given during external beam radiotherapy. The focus was to investigate the competitive relationship between cell killing and accumulation of radiation-induced DNA damage. The effects of fractionated radiation exposure at different doses/fraction on the accumulation of residual foci and DNA damage, overall cell survival, and radiosensitivity was investigated. This study highlights three main observations: excess DNA damage accumulates in cells exposed to high dosage levels; the accumulation of DNA damage PFRE is cell type-specific; and the number of fractions plays a significant role in the accumulation of DNA damage.

Cell growth data from the population doubling curves already validate the current dogma concerning the sparing effects of dose fractionation [36]; as observed, both types of cell repair sublethal damage and resume cell growth at dose levels ranging from 0.25–1.0 Gy/fraction. However, the declining slope in the population doubling curve DFRE (Figure 2A) indicates more prolonged cell cycle arrests, possibly due to DNA repair resulting from the increasing magnitude of accumulated DNA damage as dose levels increase. Persistent growth arrest in the cells irradiated at 2 Gy/fraction (Figure 2C) indicates that this is the dose level limit for sublethal damage repair in both cell types. In support of this, 3 days PFRE, excess residual γH2AX foci per cell are observed mainly in cells irradiated at 1–2 Gy/fraction. Numerous studies have shown a strong correlation between declining cell survival and increasing frequency of residual γH2AX [37,38], thus residual γH2AX foci serve as a double marker for both DNA damage and cell death (either via apoptosis or senescence).

The excess residual DNA damage in both cell types irradiated at 1–2 Gy/fraction for 3 days PFRE correlates with increased cell-killing events. Additionally, as a predictive marker for both DNA damage and cell death, the excess residual foci also highlight the dilemma at high doses: the competitive relationship between cell killing and the accumulation of genomic instability. The loss of residual γH2AX foci at doses above 10 Gy in AHH-1 seven days PFRE could be attributed to increasing depletion of apoptotic cells from the population, and this is further evident by the reduced cell growth at 1 Gy/fraction and the negative population doubling values at 2 Gy/fraction. However, the observed trend of loss of foci in VH10 fibroblasts at total absorbed doses below 10 Gy (after 9 fractions) might also indicate slower DNA repair, as this was previously shown to take weeks [39]. Temporal differences in dose–response relationships generated using residual γH2AX foci were reported earlier [37] where variations were observed in the dose–response of residual foci measured 12 and 24 h after radiation exposure. Timepoint-dependent dose–response relationships generated with residual foci suggest that this endpoint might not be a reliable marker for evaluating the risk of late detriment in cycling cells. The loss of residual γH2AX foci 7 days PFRE might indicate either the clearance of apoptotic cells or the decay of radiation-induced foci due to DNA damage repair.

The cell type-specific differences observed in this study were in the accumulation of MN, NBD and GN, the differential patterns in overall cell survival, and the changes in subsequent radiosensitivity. Radiation-induced MN formation indicates the loss of genetic information due to the loss of a chromosome or chromatid fragments during chromosomal segregation [24]. A key factor modulating the differential pattern in MN formation could be the programmed cell death (PCD) pathway in each cell type. Cell death was not investigated directly; however, it can be inferred from endpoints such as the persistent cell arrest at 2 Gy/fraction (VH10 fibroblasts), the negative population doubling at 2 Gy/fraction (AHH-1 lymphoblasts), and the loss of clonogenicity as well as the persistence of residual γH2AX foci representing unrepaired DSBs. Previous studies have shown that IR causes stress-induced premature senescence (SIPS) in VH10 fibroblasts [33], and apoptosis in AHH-1 lymphoblasts [34]. SIPS is characterized by persistent cell cycle arrest in the G1 phase due to upregulation of cyclin-dependent kinase inhibitor expression in p21 and p16INKA, unrepaired DNA damage, and the presence of senescence-associated secretory phenotype (SASP) [40,41,42]. The plateauing MN formation in VH10 fibroblasts can be attributed to the increased presence of senescent cells at higher dose levels. This raises the question, does the accumulation of MN at lower dose levels imply that the induction of SIPS has a threshold dose? If so, then cell types which rely on senescence as PCD pathway are likely to accumulate pre-malignant mutations at low dose levels. The implications are relevant for treatment modalities such as intensity-modulated radiation therapy (IMRT), where a large volume of normal tissue is exposed to low radiation doses. A higher second cancer risk was predicted early on in patients treated with IMRT than in patients treated with conventional conformal RT [15]. However, opposing results also exist, demonstrating the complexity of the problem [43].

MN formation in AHH-1 cells increased exponentially at dose levels above 10 Gy despite the increase in apoptosis (evident in negative population doubling) (Figure 4C), raising the question of whether apoptosis is less efficient than senescence in suppressing the accumulation of pre-malignant cells at high dose levels? The increase in MN formation in AHH-1 lymphoblasts could result from mitotic catastrophe at high dose levels. This is further verified by the increasing frequency of GN at high dose levels, which suggests inactivation of tumor suppressor p53 and mitotic catastrophe [30,44], since the accumulation of polyploid giant cells after irradiation in p53 mutated cells or p53^−/−^cells is well established [30,45]. Following a severe genotoxic event, these cells enter an endoreplication cycle where mitosis is entirely circumvented as a means to evade apoptosis. Polyploid cells are more likely to undergo mitotic catastrophe, which leads to increased micronuclei formation [46]. The inactivation of p53 in these cells bypasses G1 cell cycle checkpoints mediated by p53 target genes such as p21WAF1/CIP1, leading to accumulation at G2, checkpoint adaptation and aberrant mitosis yielding micronuclei [30,47]. The increasing MN frequency in AHH-1 cells is not evidence that apoptosis is a less efficient PCD pathway, but rather that multifractional radiation exposure at high doses could lead to the selection of p53-deficient cells, which might later either undergo apoptosis or necrosis in a p53-independent manner or survive with genomic instability. Therefore, multifractional radiation exposure at high dose levels might enable the selection of mutated cells in specific cell types. Our data also suggests that multifractional radiation exposure could lead to the inactivation of p53; however, unlike apoptosis in AHH-1 cells, senescence in VH10 fibroblasts could also be activated via a p53-independent pathway by direct activation of p16INKA or via RAS activation, as observed in oncogene-induced senescence [42,48].

VH10 fibroblasts experienced a dose-dependent decline in colony formation, further indicating an increasing population of senescent cells at high dose levels. The elevated cell survival of AHH-1 lymphoblasts at doses ≤ 1 Gy/fraction suggests increased cell proliferation, which is required for cell repopulation (Figure 6A). The rationale behind cell repopulation is the replacement of lost cells, which is characteristic of this cell type. Although not shown, we observed an increase in the growth rate of irradiated AHH-1 cells towards the end of the experiment, but only in cells irradiated at dose levels ≤ 1 Gy/fraction. AHH-1 cells irradiated at 2 Gy/fraction had the lowest colony-forming ability, indicating a change in cell growth characteristics. Another possible explanation for the decline in colony formation in AHH-1 cells irradiated at 2 Gy/fraction could be delayed apoptosis or mitotic problems due to the high frequency of GN, or a switch to a more senescent-like state.

The dose-dependent increase in AHH-1 radioresistance further suggests the inactivation of p53 in AHH-1 cells. The increased cellular radioresistance can be attributed to a number of factors, one of which is apoptosis resistance [49]. Cancer cells with mutant p53 have been shown to evade apoptosis via the expression of anti-apoptotic genes within the BCL family, and the suppression of pro-apoptotic genes BAX, PUMA (BBC3) and NOXA (PMAIP1) [42,48,50]. Apoptosis resistance also has been attributed to expression of microRNAs. such as miR-34a in head and neck cancer [51], or the expression of activating transcription factor 3 (ATF3) in breast cancer cell lines [49]. Improved DNA repair capacity also increases the cellular radioresistance, as shown in lung cancer extracellular signal-regulated kinase 5 (ERK5), also known as MAPK7, which improved DNA repair via the homologous recombination repair pathway via the activation of Chk1 [52]. Additionally, an improved antioxidant response could increase the radioresistance. The expression of NRF2 (NFE2L2) transcription factor has been shown to be activated as a delayed response to ionizing radiation both after a single acute high dose or daily fractions [53]. In p53-mutated cancer cells, NRF2 activation of the antioxidant response pathway is a pro-survival mechanism that is constitutively active, which could play a role in the radioresistance of cancer cells [54]. In addition, NRF2 activation is also strongly associated with the expression of stem cell markers [55]. EMT-like phenotypic transitions which transform differentiated cells to dedifferentiated cells with stem cell-like properties is related to increased expression of certain normal and cancer stem cell markers such as Slug, Twist, Sox2, CD44, Nanog, CD133 (PROM1), Oct4, and ABC membrane transporters. Additionally, this process is linked to the activation of the Hedgehog pathway and upregulation of the Wnt/β-catenin and downregulation of Notch signaling pathway, and have long been associated with increased the radioresistance and chemoresistance in cancer cells [56], while less is known for normal cells. The differences in effects observed in VH10 fibroblasts after 9 and 12 fractions include lower focus frequencies at all dose levels, lower MN frequency at dose levels below 1 Gy/fraction, and better colony-forming potential in VH10 cells after 12 fractions. Increasing the number of fractions at dose levels below 1 Gy/fraction might play a role in cell killing at these dose levels. A switch in cell death pathways from senescence to apoptosis is highlighted by the depletion of damaged cells with MN at these doses, and the improved cell survival of VH10 cells after 12 fractions. The decision to switch to apoptosis in normal fibroblasts is modulated by low p21 expression [57] or posttranslational modifications such as acetylation of lysine 117 on p53 [48], while phosphorylation of serine-15 and threonine-18 are associated with senescence [58]. This reduced biological effect suggests apoptosis might be activated at low dose levels by increasing number of fractions.

The results of this study suggest a possible correlation between cell-dependent factors and the shape of the curve predicted by risk models. Thus, tissues with cells prone to senescence might show competition between the induction of mutations and cell killing with bell-shaped risk curves, while cells prone to apoptosis might have a plateau risk model consistent with accumulation of damage at high doses. This hypothesis could be validated in the future provided that epidemiological studies will be designed to consider the type of cells characterized from the point of view of the dominant cell-killing mechanism. These results also indicate that fractionation plays a role for the accumulation of mutations and hence should be considered for modelling purposes. Differentiating between different types of cells, in the context of the fractionated exposure to doses in the range expected at the periphery of the radiotherapy fields, might therefore lead to further development of more accurate models for predicting the risk posed to SMN.

Limitations of this study include the time points chosen. Three and seven days PFRE might be too late as time points for analyzing the effects of multifractional radiation exposure on the accumulation of DNA lesions. This is particularly evident in AHH-1 cells, which divide rapidly. Thus, we did not see any changes in MN and GN frequency at low dose levels at these time points. It is possible that a higher frequency of these effects could be detected at an earlier time point (24–48 h after last fraction). The low residual foci frequency seen generally in VH10 fibroblasts after 12 fractions could be affected by technical issues. However, we observed a similar decline in MN and NBD frequency in cells after 12 fractions. Clonogenic survival assays, assaying the radiosensitivity of VH10 fibroblasts after 12 fractions, were not included due to technical problems. The analysis of MN, NBD and GN alongside with residual DNA damage was circumstantial but has provided interesting insights as well as posed new questions. We are now interested in determining if the dose–response relationship observed with MN frequency will be similar to that of radiation-induced translocations. If that is not the case, then other factors such as survival of the cells treated at 2 Gy/fraction might play a role in the discrepancy in shape of the dose–response curve. Further studies will also include in-depth mechanistic analysis to better understand cell-specific modulators involved in the accumulation of DNA lesions.

## 4. Materials and Methods

### 4.1. Cell Culture

Normal human foreskin fibroblasts (VH10) donated by Leiden University were cultured in Dulbecco’s modified minimum essential medium (DMEM, Sigma-Aldrich, Schnelldorf, Germany), supplemented with 10% bovine calf serum (HyClone, Thermo Fisher Scientific, Waltham, MA, USA) and 1% penicillin-streptomycin (10,000 U penicillin and 10 mg streptomycin/mL, Sigma-Aldrich). All experiments started with fibroblasts at passage 7 (P7) grown to 80% confluence before the start of each experiment. VH10 fibroblasts were passaged weekly in 75-cm^2^ flasks at a seeding density of 3.5 × 10^5^ cells. AHH-1 lymphoblasts (ATCC, USA, cat nr. CRL-8146) were cultured in RPMI-1640 medium with 25 mM HEPES (Sigma-Aldrich), supplemented with 10% bovine calf serum (HyClone), 1% penicillin-streptomycin, 1% L-glutamine (200 mM), and 1% sodium pyruvate solution (100 mM), all from Sigma-Aldrich. AHH-1 lymphoblasts were passaged three times weekly in 75-cm^2^ culture flasks at a seeding density of 3.0 × 10^6^ cells during weekdays and 2.0 × 10^6^ cells during weekends. All the cells were grown at 37 °C and 5% CO_2_.

### 4.2. Multi-Fraction Radiation Exposure

Cells were exposed repeatedly to gamma radiation from a ^137^Cs source Gamma Cell^®^ Exactor 40 (Best Theratronics, Ottawa, ON, Canada) at room temperature, at 0.77 Gy/min. Non-irradiated control samples were sham exposed. The radiation exposure of VH10 fibroblasts always started on a Monday, followed by exposures on Wednesday and Friday (3 fractions per week); VH10 fibroblasts were irradiated for a total of 3 weeks (21 days), and passaged once a week. Multifractional radiation exposure of AHH-1 cells always started on a Monday and was followed by exposures on Tuesday–Friday (5 fractions per week), AHH-1 cells were irradiated for two weeks, and passaged three times a week (Figure 1).

### 4.3. Cell Growth, Colony Formation Assay and Assessment of Radiosensitivity by Clonogenic Survival Assay

Cell density and viability were determined during passaging, using the trypan blue exclusion assay, as previously described [59], using an automated cell counter (Cell Countess, Paisely, PA, USA, Invitrogen, Paisley, UK). Colony-forming assays of cells repeatedly exposed to gamma radiation were performed 10 days after the last fraction, using the agarose overlay colony formation assay as described in [60] for VH10 fibroblasts, or by the soft agar colony-forming assay as described in [61] for AHH-1 lymphoblasts. The radiosensitivity of VH10 fibroblasts, previously exposed to fractionated irradiation with a new dose, was assessed using the assay described above, where cells were seeded at a density of 5000 cells in 100 mm diameter cell culture dishes, irradiated 24 h later at 2–8 Gy, and then incubated for 21 days. For radiosensitivity assessment of AHH-1 lymphoblasts, cells were seeded at cell densities ranging from 1000–4000 cells in duplicates in six-well culture dishes, irradiated at 1–4 Gy and incubated for 10 days.

### 4.4. γH2AX Immunofluorescence Assay

Six hours after the last fraction of radiation, VH10 fibroblasts were detached using trypsin and seeded at high density in duplicates on 22 mm × 22 mm coverslips (VWR International, Sweden), placed in six-well plates containing medium, and incubated at 37 °C for either 3 or 7 days. After incubation, cells were fixed with 70% ethanol for 10 min at room temperature. Three or seven days post-fractionated radiation exposure 1 × 10^5^ AHH-1 lymphoblast cells were attached to polylysine-coated slides via cyto-centrifugation at 800 rpm for 5 min and air-dried for 1–2 min. AHH-l lymphoblasts were fixed in 3% paraformaldehyde and 2% sucrose in PBS for 15 min, followed by two PBS washes. Immunostaining was carried out as described in [62].

### 4.5. Image Acquisition and Analysis of γH2AX Foci

Individual cells were selected randomly and captured using a fluorescent microscope with a 100× oil immersion lens (Nikon Eclipse E800, Nikon, Tokyo, Japan), and the image analysis system ISIS (Metasystems, Althusheim, Germany) coupled to a CCD camera [50]. A modified macro written for ImageJ software [37], version 1.43u, was used to calculate the area and number of γH2AX foci. Foci were categorized as either small (8 and 60 pixels) or large (61–500 pixels) foci based on their areas as described in [62]. A total of 100 cells were analyzed for each dose per experiment.

### 4.6. Micronuclei, Nuclear Buds, and Giant Nuclei

Micronuclei (MN), nuclear buds (NBD), and giant nuclei GN) were scored on the same images as γH2AX foci. MN are small nuclear bodies lying close to, but not connected to the main nucleus, while NBD are connected to the nucleus via a stalk. MN and NBD frequency was scored in mononucleated cells using the standard criteria [63]. Using a function of ISIS, giant nuclei (GN) were identified by their area, which ranged from 80–160 μm^2^, compared to the control nuclei which had an average nuclei area of 35 μm^2^.

### 4.7. Statistical Analysis and Data Fitting

Multiple comparisons were carried out using either one- or two-way ANOVA (GraphPad Prism ver. 9.3.1), and multiple comparisons were corrected using Tukey’s post hoc test. A *p* value below 0.05 was considered significant. Cell growth curves were fitted to a second-order polynomial equation. Residual γH2AX foci and MN, NBD, and GN were fitted to either a linear or linear quadratic equation. Simulations of MN, NBD frequencies were carried out using Python 3 software (ver. 3.7.9, Python Software Foundation, Wilmington, DE, USA). Survival curves for fractionated radiation exposure were fitted using the linear quadratic equation $S=e^{-\left(D\left(a+\beta d\right)\right)}$, where *D* is the total absorbed radiation dose in Gy, *α* and *β* are fitting coefficients, and d is the dose in one fraction. Survival curves of VH10 fibroblasts, reradiated 20 days PFRE, were fitted using the linear quadratic equation $\;S=e^{-\left(aD+\beta D²\right)}$ where *D* is the dose in Gy, *α* and *β* are fitting coefficients. AHH-1 lymphoblasts were fitted to a linear equation $S=e^{-\left(\alpha D\right)}$ because AHH-1 lymphoblasts are radiosensitive fast-dividing cells.

## 5. Conclusions

In conclusion, this study highlights an excess DNA damage in cells exposed to fractionated high doses of radiation. Additionally, cell type-specific factors modulate the accumulation of DNA lesions PFRE, where cell death pathways are indicated in particular. The fractionation scheme also plays a significant role in the accumulation of radiation-induced DNA lesions, as suggested by the differences seen between the number of fractions in VH10 fibroblasts.

## Figures and Tables

**Figure 1 ijms-23-12861-f001:**
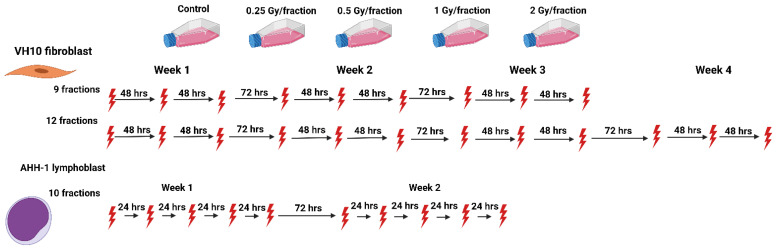
Schematic representation of the fractionation schemes. The red thunder icon represents individual radiation fractions created with Biorender.com.

**Figure 2 ijms-23-12861-f002:**
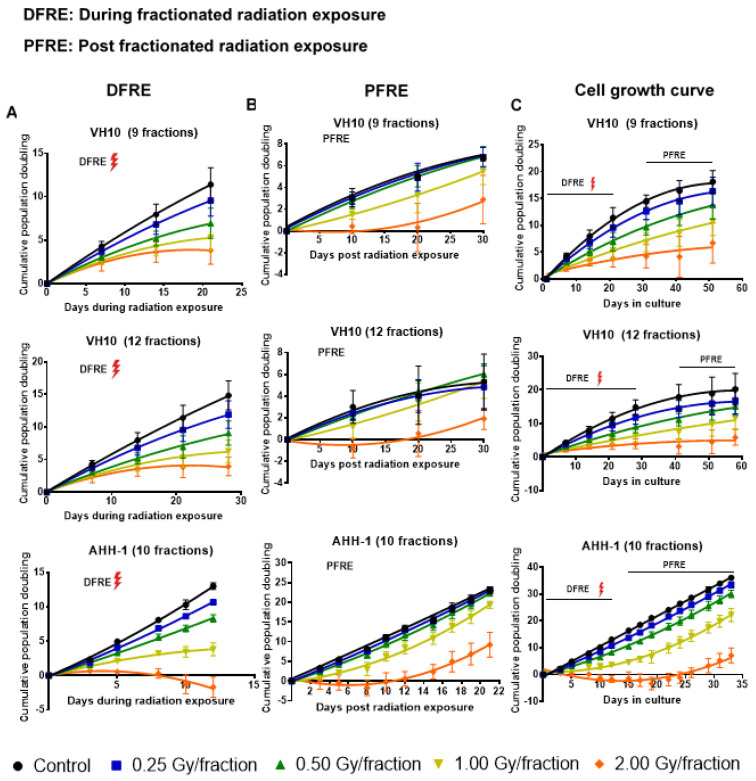
Cell growth curves. (**A**) Cumulative population doubling curves of cell growth during fractionated radiation exposure (DFRE). (**B**) Cumulative population doubling curves of cell growth post-fractionated radiation exposure (PFRE). (**C**) Cumulative population doubling curves of cell growth from the start of the experiment to the end. The red thunder icon represents radiation exposure. Cell growth data was fitted to a second-order polynomial equation. B0, B1 and B2 coefficients were summarized in Appendix A. Average B1 coefficients of the cell growth curves from different dose levels were compared using one-way ANOVA and corrected with Tukey’s post hoc test. Adjusted *p* values and 95% confidence intervals are summarized in Appendix A. Error bars represent standard deviations.

**Figure 3 ijms-23-12861-f003:**
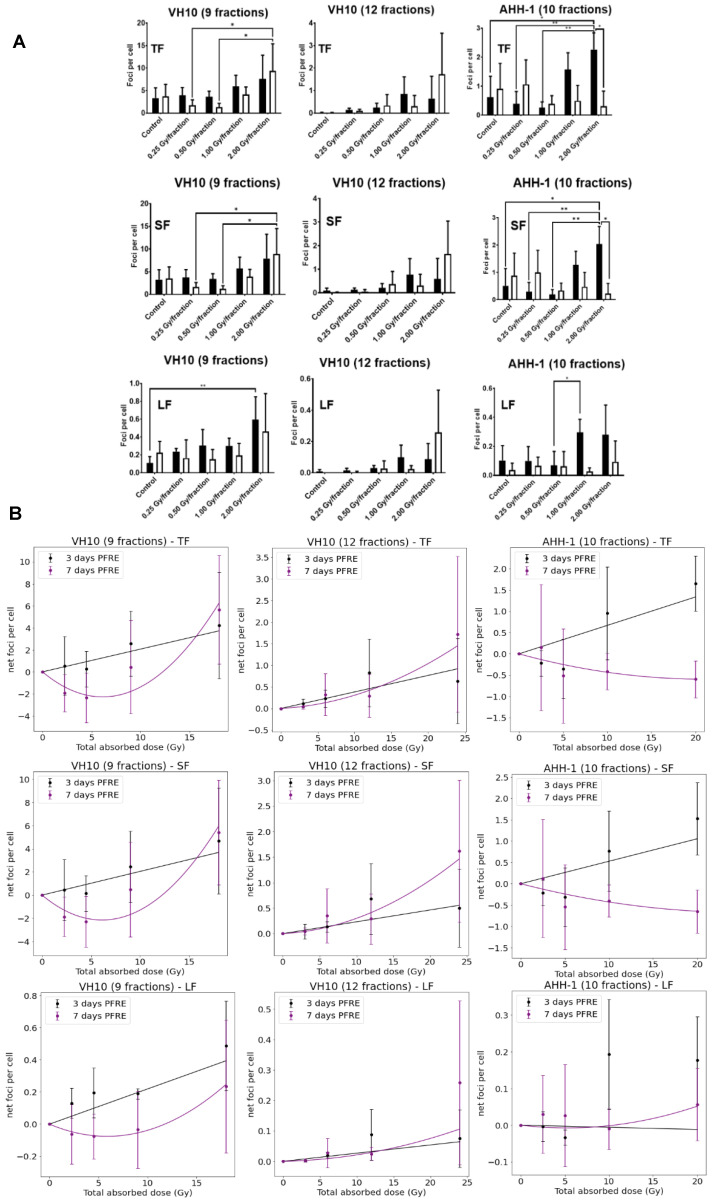
Residual DNA damage post-fractionated radiation at different dose levels. (**A**) Accumulation of residual DNA damage in the form of γH2AX foci at different dose levels per fraction. Black bars represent three days post-fractionated radiation exposure; white bars represent seven days post-fractionated radiation. * represents *p* values < 0.05, ** represents *p* values < 0.01. (**B**) Dose–response curves of residual foci 3 and 7 days post-fractionated radiation exposure as a function of total absorbed dose. TF represents total foci, SF represents small foci, LF represents large foci. Data were normalized by subtracting the values of the non-irradiated control from the irradiated samples. Net foci values three days post-radiation exposure best fit a linear relationship, while net foci values seven days post-fractionated radiation exposure included a linear quadratic equation. Data fitting coefficients were summarized in Appendix A. Three independent experiments were performed for VH10 fibroblasts, while four independent experiments were performed for AHH-1 lymphoblasts.

**Figure 4 ijms-23-12861-f004:**
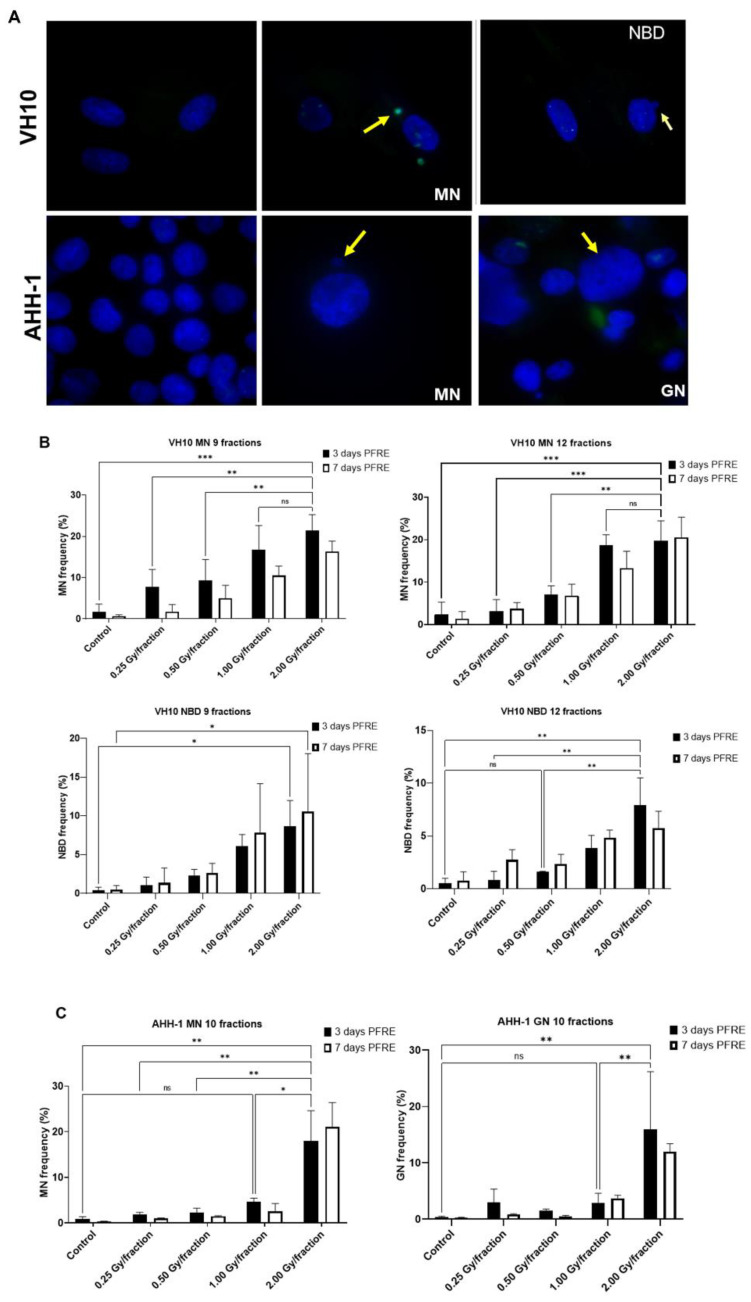
Accumulation of markers of genomic instability: Micronuclei MN, nuclear buds NBD and giant nuclei GN in cells post-fractionated radiation exposure. (**A**) Fluorescence images of nuclei with micronuclei MN, nuclear buds NBD, and giant nuclei in both cell types. Red arrows indicate MN, while yellow arrows indicate the NBD and GN. Red arrows indicate micronuclei. Yellow arrows indicate nuclear buds in the VH10 panel and giant nuclei in the AHH-1 panel. (**B**) Percentage of cells with micronuclei MN and nuclear buds NBD in VH10 fibroblasts. (**C**) Percentage of cells with MN and giant nuclei 3 days and 7 days post-fractionated radiation exposure. ns represents no significant difference * represent *p* values < 0.05, ** represents *p* values < 0.01, *** represents *p* values < 0.001.

**Figure 5 ijms-23-12861-f005:**
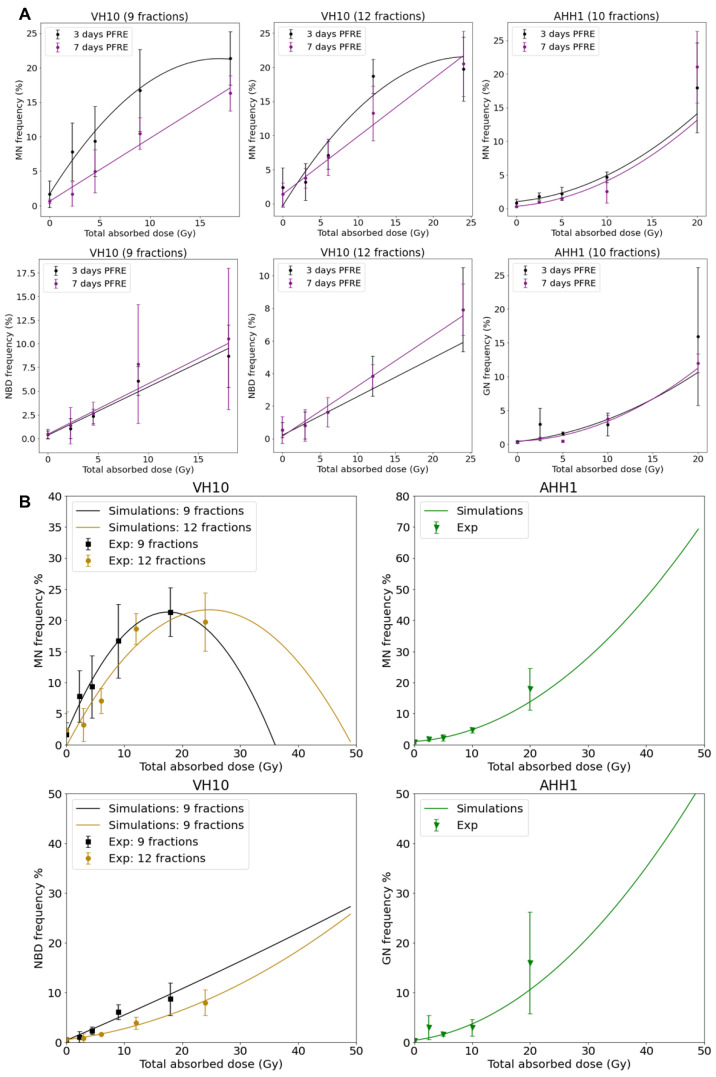
Dose–response relationships as a function of total absorbed dose for each cell type. (**A**) Dose–response relationship was plotted using MN, NBD, and GN frequency, and ultimately obtained 3 and 7 days PFRE as a function of total absorbed dose. Data fitting coefficients are summarized in Appendix A. MN and GN were fit to a linear quadratic function, while NBD were fitted to a linear function. Three independent experiments were performed for both VH10 fibroblasts and AHH-1 lymphoblasts. (**B**) Generated simple simulations using the data fitting coefficients were obtained 3 days PFRE (Appendix A). In these simulations, the total absorbed dose is extended to 50 Gy.

**Figure 6 ijms-23-12861-f006:**
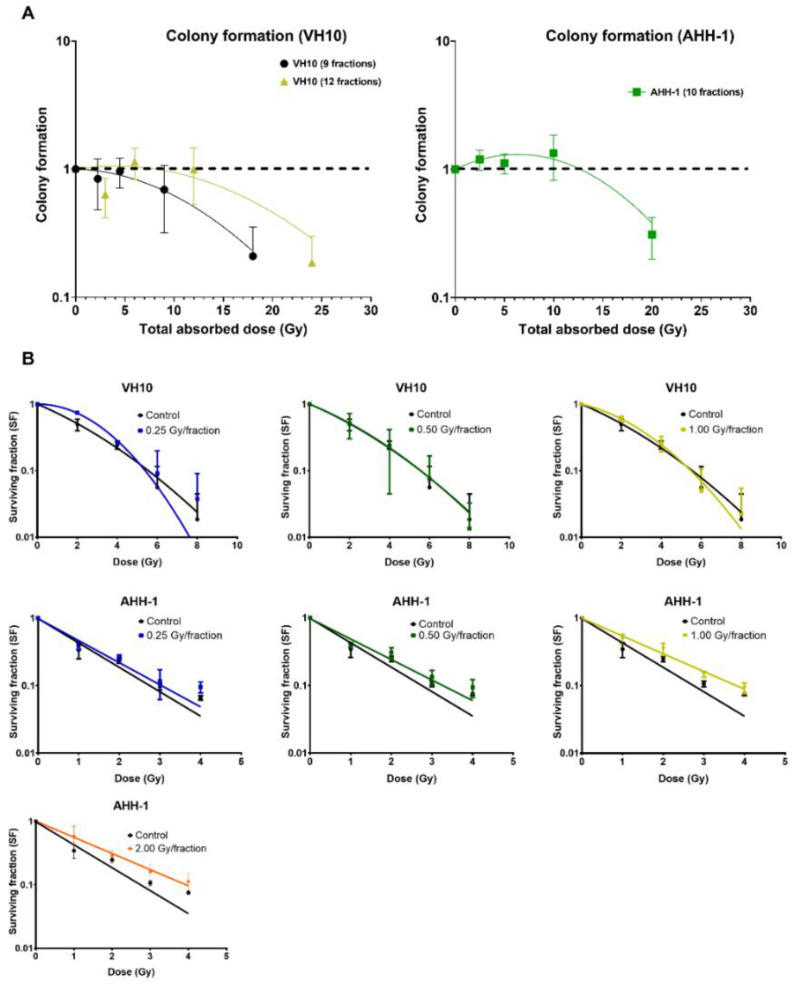
Differential patterns in cell survival and radiosensitivity: (**A**) Overall cell survival using colony-forming assays were carried out 10 days post-fractionated radiation exposure. Data was plot as a function of the total absorbed dose. Survival curves were fitted to a linear–quadratic equation. Three independent experiments were performed for VH10 fibroblasts, while four independent experiments were performed for AHH-1 cells. Summary of α and β coefficients for survival curves are given in Appendix A. (**B**) Radiosensitivity of VH10 and AHH-1 cells, when challenged 20 days post-fractionated radiation exposure with a single acute dose. Three independent experiments were performed for VH10, and four independent experiments were performed for AHH-1 lymphoblasts. Summary of α and β coefficients for VH10 survival curves in Appendix A.

**Table 1 ijms-23-12861-t001:** Summary of total absorbed dose at each dose level.

Total Absorbed Doses				
	0.25 Gy/Fraction	0.5 Gy/Fraction	1.0 Gy/Fraction	2.0 Gy/Fraction
VH10 (9 fractions)	2.25 Gy	4.5 Gy	9.0 Gy	18.0 Gy
VH10 (12 fractions)	3.0 Gy	6.0 Gy	12.0 Gy	24.0 Gy
AHH-1 (10 fractions)	2.5 Gy	5.0 Gy	10 Gy	20.0 Gy

## Data Availability

The data presented in this study are available on request from the corresponding author.

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
