# Peer review of "Cell Type-Specific Patterns in the Accumulation of DNA Damage Following Multifractional Radiation Exposure"

_ijms, 2022, doi:10.3390/ijms232112861_

Round 1

Reviewer 1 Report

This study discussed the correlation between cell killing, accumulation of DNA lesions. The author compared during 126 fractionated radiation exposure and post fractionated radiation exposure in different cell types. The results showed that the competitive relationship between cell killing and the accumulation of carcinogenic DNA damage following multifractional radiation exposure is cell-type-specific. Some advinces need to let authors know. 

1. The manuscript needs to be edited to clear the grammatical errors. such as CO"2" or 1×10"5" AHH-1 lympho-478 blast cells

2.  In figure 3, "white bars represent seven days post fractionated radiation. * 163 represent p values >0.05, ** represents p values >0.01.", is it right?

3. In table S4-S7, are the authors sure that "," is right ?

4. In table S8, the data "p-value (R 2 test)" is p-value or 2, which one?

5. All figures of survival fraction need unite. Some data showed lines, and some data showed curves. 

Reviewer 2 Report

1. In the introduction section, paragraph 1 and 2, the references are too old. In fact, journal of clinical oncology had already drawn a line about predictive secondary neoplasms after radiation therapy. It’s a linear form except thyroid cancer. This is from the long term followed up of “Childhood Cancer Survivor Study Cohort”. Authors might search this study. J Clin Oncol. 2009 May 10;27(14):2356-62. This issue had been well described.

2. There are many articles discuss this issue. For example, “Effect of Low Dose Rate Irradiation on the Division Potential of Cells in vitro. VII. Human Fibroblasts from Young and Adult Donors. Comparative Study In Vitro. 1980 Jul;16(7):634-8. “

3. There are already cell culture model in fibroblasts, lymphoblast, e.g., “Preclinical models in radiation oncology. Radiat Oncol . 2012 Dec 27;7:223.” review article. I wonder what’s new in this article.

4. Also, 3d tissue models as tools had been proposed, e.g., 3d tissue models as tools for radiotherapy screening for pancreatic cancer., Br J Radiol. 2021 Apr 1;94(1120):20201397.

5. The radiation-induced genomic instability and bystander effects had been also discussed widely.

Overall, not very new could obtain from this article.

Minor

1. The references are too old. Page 1, line 36,  “… contributed to a steady rise in the 5-year survival of cancer patients [1]”. The reference year is 2012.

2. Line 37” treat ca 50%” ca should be cancer.

Reviewer 3 Report

The presented paper uncovers an in vitro study to investigate the biological effect of heterogeneous dose distribution on the competition between cell killing and accumulation of DNA damage in surviving cells. The authors designed fractionation schemes tailored to two human normal cell types (VH10 fibroblasts and AHH-1 lymphoblasts) representing elastic and hierarchical tissues. The presented study highlights an excess DNA damage in cells exposed to fractionated high doses of radiation. The results obtained by the authors extend the experimental results of low-dose fractionation.

The study design is appropriate for the its aim. However, the Introduction and Discussion sections should be extended by some of latest papers on cellular radioresistance:

- Multifractional radiation exposure and Hypoxia-induced cell responses [Kabakov, A. E., & Yakimova, A. O. (2021). Hypoxia-induced cancer cell responses driving radioresistance of hypoxic tumors: Approaches to targeting and radiosensitizing. Cancers, 13(5), 1102.]

- Multifractional radiation exposure and Molecular Chaperones [Kabakov, A., Yakimova, A., & Matchuk, O. (2020). Molecular chaperones in cancer stem cells: determinants of stemness and potential targets for antitumor therapy. Cells, 9(4), 892.]

- Mechanisms of radiation resistance in cancer cells [Alamilla-Presuel, J. C., Burgos-Molina, A. M., González-Vidal, A., Sendra-Portero, F., & Ruiz-Gómez, M. J. (2022). Factors and molecular mechanisms of radiation resistance in cancer cells. International Journal of Radiation Biology, 1-15.]

Round 2

Reviewer 1 Report

Thanks for your revision. However, in supplementary data S4-S6, the authors should define the significance *, **, ***, and ****. Please identify the P value in the figure (table) legend.

Reviewer 2 Report

1. Review article regarding mechanisms of injury of normal tissue after radiotherapy has been published.  E.g. “Mechanisms of injury to normal tissue after radiotherapy: a review. PMID: 24374687”

e.g. Hoejmakers JHJ. DNA Damage, Aging, and Cancer. N Engl J Med. 2009; 361(15):1475–1485.[PubMed: 19812404]; Zhao W, Diz DI, Robbins ME. Oxidative damage pathways in relation to normal tissue injury.British J of Rad. 2007; 80:S23–S31.; Lipinski B. Hydroxyl Radical and Its Scavengers in Health and Disease. Oxidative Medicine and Cellular Longevity. 2011 doi:10.1155/2011/809696.; Dotto GP. p21WAF1/Cip1: more than a break to the cell cycle. Biochimica et Biophysica. 2000;

1471:M43–M56; Strausfeld U, Labbé JC, Fesquet D, Cavadore JC, Picard A, Sadhu K, Russell P, Dorée M.Dephosphorylation and activation of a p34cdc2/cyclin B complex in vitro by human CDC25protein. Nature. 1991; 351:242–245. [PubMed: 1828290]

; Huang X, Halicka HD, Darzynkiewicz Z. Detection of Histone H2AX Phosphorylation on Ser-139 as an Indicator of DNA Damage. Current Protocols in Cytometry. 2004:7.27.1–7.27.7. [PubMed:18770792]

2. Also, DNA damages and repair mechanisms are well known. DNA damage didn’t equal to carcinogenesis.  We cannot get the conclusions merely from cell culture. I think it’s too early to bold to jump from DNA damage and then carcinogenesis conclusion. One didn’t know which mechanism predominate and lead to cancer.

3. What is “MN” appeared first in page 6 of 19, line 206? I am supposed it is “micronuclei

4. Radiation induced micronuclei, nuclear buds, mononucleated nuclei in normal cell are well known. But these lead to cancer are too earlier conclusion and that’s why I think it’s important to get some initial impressed from 3D model.

5. the short abbreviation should be state first instead of assuming everyone known. For example version 1, “treat ca 50%”. And the authors are still not tried to correct it.
